*Research Directions: Depression*

# Keeping time: investigating the relationship between circadian rhythm and mood disorders via precise circadian measurement techniques

Katherine M. Lyman and Colleen A. McClung 

University of Pittsburgh School of Medicine, Department of Psychiatry, Pittsburgh, PA, USA

## Impact Paper

Circadian rhythms; sleep; mood

**Corresponding author:**
Colleen A. McClung;
Email: mcclungca@upmc.edu

## Abstract

Disruptions in circadian rhythms and sleep have long been associated with mood disorders. In fact, sleep disturbances are one of the key features used in the *Diagnostic and Statistical Manual of Mental Disorders, Fifth Edition* (DSM-V-TR) diagnosis of major depressive disorder and bipolar disorder. Sleep/wake abnormalities can also precede mood episodes and predict treatment response. Thus, precise measurement of specific sleep/circadian features is important as these measures can be used clinically to direct appropriate treatments. These measures can also be used for research purposes to try to understand specific mechanisms by which circadian rhythm disturbances and sleep/wake perturbations may lead to specific phenotypes. The purpose of this review is to highlight recent advances in methodology which can be used to more precisely measure sleep/circadian biology. This review will examine how these new methodologies can better elucidate the mechanisms linking sleep/circadian disruptions and mood disorders, as well as how new technologies can be used therapeutically to treat sleep/circadian abnormalities.

## Introduction

Sleep and circadian rhythm disturbances (SCRD) have long been implicated as a component of psychiatric pathology, with sleep disturbances noted as a symptom of many psychiatric disorders, including schizophrenia (Kaskie et al., 2017), mood disorders (American Psychiatric Association, 2013), obsessive compulsive disorder (Segalàs et al., 2021), and substance use disorders (Roehrs and Roth, 2015). Psychiatric disorders are also associated with circadian rhythm disturbance; for example, evening chronotype, in which individuals demonstrate a preference for activity in the evening rather than morning, has been associated with depression, substance use disorders, and eating disorders, and is associated with more severe mood symptoms for individuals with bipolar disorder (Zou et al., 2022). Individuals with major depressive disorder exhibit lower amplitude circadian rhythms (Kang et al., 2024), which have largely been assessed in humans using actigraphy (Lyall et al., 2018 ). Gene expression analysis of human postmortem brain tissue has also revealed weaker circadian patterns in subjects with depression when compared with controls (Li et al., 2013).

While SCRD may first become visible to patients as apparent symptoms of psychiatric illness, research has suggested a bidirectional causative role, with SCRD in fact contributing to the pathological processes of these illnesses. SCRD have been identified as risk factors in the development of bipolar disorder (Scott et al., 2022), major depressive disorder (Murphy and Peterson, 2015), schizophrenia (Waite et al., 2020), substance use disorders (Hasler and McClung, 2021), and other diagnoses.

However, investigation of sleep and circadian disturbance within human subjects has been limited due to technological constraints, with few tools available to obtain accurate, continuous rhythmic data. Polysomnography (PSG) and actigraphy play primary roles in the study of sleep patterns, although they face some limitations. Polysomnography scoring is still performed in part by individual scorers and can be subjective; additionally, participants receiving PSG are susceptible to the "first night effect" of sleeping in an unfamiliar sleep lab, and consequently producing sleep data which may not be representative of a typical night's sleep. Actigraphy relies primarily on movement data and may also give inaccurate information, often overestimating sleep periods by labeling waking periods of rest or inactivity as sleep (Trust, 2022).

Circadian rhythms are likewise difficult to study. While sleep behavior (assessed via PSG, actigraphy, or self-report) is often seen as a proxy for circadian rhythm, the true scope of circadian timekeeping extends to multiple systems beyond the sleep/wake mechanism, and originates at a molecular level which could ideally be measured using precise biomarkers. However, tools for obtaining this data have been limited.

Some biomarkers for circadian rhythm have been used previously, with varying success. Melatonin-based measure, such as salivary melatonin used to estimate dim light melatonin onset (DLMO) and urinary 6-sulfatoxymelatonin, are convenient to obtain and provide

information about the circadian phase of the subject (Crowley et al., 2016). However, both methods are susceptible to a "masking effect" in which light exposure may affect levels during sampling, and protocols around these biomarkers should be carefully designed to avoid this effect. Such a protocol design is possible but highly restrictive, as for example the Constant Routine (CR) protocol (Minors and Waterhouse, 1984), in which subjects are exposed to constant dim light and behavior is strictly controlled (i.e. via constant semi-recumbent posture, isocaloric food intake, and continued wakefulness). Core body temperature and plasma cortisol levels have also been used as biomarkers to inform estimates of circadian phase, although they have been less reliable when compared to melatonin markers (Klerman et al., 2002).

In order to more precisely understand the circadian profile of individual subjects, some innovation of circadian measurement tools is required. In the following review, we will examine the emergence of new circadian data collection technology in recent years, as well as novel protocols which have adapted existing technologies to produce more accurate, reliable, and easily obtainable data. We will examine the current state of clinical research which seeks to establish causal links between SCRD and mood disorders, and we will also review existing and emerging circadian-based therapies which appear promising in the field.

## Novel methods of circadian rhythm measurement

Numerous methodologies have emerged in recent years which allow more precise, continuous measurement of circadian rhythm using genetic, metabolic, and immunological markers. In this section we will review some of these emerging methodologies which may present new solutions to problems of circadian measurement in research design.

The unique strengths and weaknesses of circadian measurement techniques must be considered with respect to the specific knowledge goals of the study, as well as logistical considerations, cost, and the accuracy and precision possible via various techniques. Circadian data is remarkably versatile in how it may be obtained, as circadian mechanisms may be inspected within every cell of the human body across tissue type. This leaves researchers with the task of selecting the optimal circadian measurement tool from many potential options.

For example, gene expression data may be collected serially (as discussed below) in order to quickly obtain vast amounts of information about circadian transcriptional fluctuations in a group of subjects. However, this method is tissue-specific; circadian gene expression data obtained from peripheral blood *only* tells a story of gene expression within the blood, as this process is locally regulated and may appear entirely differently within the brain. As the brain cannot be accessed for serial tissue collection in living subjects, this method is ultimately limited in terms of the brain-specific continuous data measurement that would be of interest in psychiatric research.

Metabolic methods are also discussed below; the circadian nature of metabolism has been studied across discipline, with endocrinology and kinesiology research also exploring new methods of circadian measurement in this arena. There have been remarkable technological advances in the study of metabolism, with biosensors now showing promise for the collection of continuous biodata along numerous parameters. While these technologies measure peripheral markers, some of these markers can also reflect rhythmic patterns in the brain (i.e. cortisol, which connects peripheral and central systems via the hypothalamic-

pituitary-gonadal axis), consequently yielding potential usefulness for psychiatric research.

Notably, for many of the methodologies discussed below, there is currently a lack of extensive data verifying their utility, as they involve novel techniques and technologies which have yet to be tried in multiple research settings and study designs. However, we present these methodologies here as a review of possibilities for future circadian research, which may become more proven with time.

## Circadian patterns of gene expression

Gene expression patterns within various cell types are known to vary throughout the 24-hour cycle, and may provide a rich potential source of individual circadian data (Takahashi, 2017); however, obtaining gene expression data from human subjects around a 24-hour cycle poses challenges in feasibility. Here we will review recent studies which have innovated existing techniques in order to produce accurate circadian transcriptional data.

### Oral sampling

Gene expression data may be painlessly obtained orally from human subjects; when collected serially over multiple timepoints, this data may be used to track transcriptional patterns of individuals over the 24-hour cycle. For example, in 2024, Das et al. collected serial saliva samples from children with fetal alcohol syndrome in order to study the unique circadian expression of core-clock genes and their regulatory genes in this disorder. In 2022, Vasko et al. utilized buccal epithelium sampling in order to assess circadian variation of Per1, Clock, Bmal1, and Cry1 genes in healthy subjects with different chronotypes, finding that those with an evening chronotype had a higher evening expression of Clock.

Despite the convenience of this method for study subjects, oral sampling is notably confounded by a high bacterial RNA presence (Kumar et al., 2023). Indeed, the fraction of human RNA in saliva is generally low due to bacterial contamination, thus compromising the quality of genetic data obtained via this method. In 2022, Ostheim et al. developed a novel protocol to address this issue, using poly(A)+-tail primers followed by qRT-PCR in order to select for cDNA synthesis for human RNA species (Ostheim et al., 2022). Use of protocols such as this one may improve the accuracy of oral sampling in further circadian transcriptional studies.

### Blood sampling

While an older method, serial blood sampling is still used to collect circadian transcriptional data from study subjects by measuring gene expression patterns within peripheral blood cells at different points in time. Circadian disruption in the periphery has a noted correlation with SCRD of central origin. Studying peripheral blood cell gene expression can be useful for understanding the full scope of circadian dysfunction within psychiatric disorders, and how this disruption affects multiple organ systems. For example, Koritala et al. described circadian disturbances seen in circulating leukocytes in individuals with a night shift schedule (Koritala et al., 2021). Tu et al. also recently studied peripheral blood gene expression for patients with Parkinson's disease (PD), a neurological disease known to alter sleep patterns. Their findings noted an altered rhythm of autophagy in PD when compared to controls, raising questions about the transsystemic reach of central circadian disruption and the mechanisms by which distantly related biomarkers may be implicated in centrally-originating SCRD (Tu et al., 2021).

### Cross-tissue analysis

Further research in circadian gene expression may be informed by the Genotype-Tissue Expression (GTex) project of the Broad Institute, which provides comprehensive data on gene expression across tissue types. These samples are obtained from tissue samples donated by subjects via postmortem autopsy or transplant settings (Broad Institute of MIT and Harvard, 2025). Talamanca et al. (Talamanca et al., 2023) developed an algorithm to study circadian patterns within the GTex dataset, finding conserved timing of clock transcripts throughout the body. However, they also noted that circadian patterns were highly sex-dimorphic and that rhythms generally dampened with age. These findings may be useful when designing future studies which rely on peripheral markers of circadian gene expression. In particular, the tight conservation of these rhythms supports more generalizable inferences regarding SCRD from the sampling of a variety of tissue types; at the same time, sex and age differences should be accounted for in order to ensure accurate interpretation of results.

### Circadian patterns of metabolism and immune function

Metabolism and immune function both follow a 24-hour pattern and have long been studied as circadian entities. In the past, this has been done by tracking cortisol levels (which impact both metabolic and immune function), or by tracking core body temperature as a proxy of metabolic activity. In recent years, research in this domain has transformed through the development of new technologies which allow precise 24-hour data collection of various metabolic biomarkers. Many of these technologies have yet to be used in psychiatric research, and a multidisciplinary approach may be required in order to better utilize these techniques in psychiatric circadian science.

### Advancements in actigraphy

Actigraphy has often been used as a proxy for metabolic activity, as periods of rest and inactivity constitute one dimension of an individual's metabolic output. However, recent developments in actigraphy have added to the usefulness of this tool. Multimodal wearable sensing, such as that obtainable through Fitbit data, allows researchers to combine multiple metabolic parameters (heart rate, step count, and rest-activity data) in order to create a more accurate metabolic proxy of circadian rhythm (Zhang et al., 2024). Standard actigraphy has also improved through the development of new mathematical models which improve its ability to predict circadian phase. Moreno et al. developed such a model which showed improved accuracy when compared to the standard DLMO method (Moreno et al., 2022). Berlin et al. also developed an "octagonal actigraph" which combined collected continuous measurements from the hip and wrist and combined this data with heart rate measurements; they found that their data was comparable to oxygen consumption values in terms of accurately measuring metabolic rates over time (Berlin et al., 2007).

### Biosensors

New biosensors have become available which allow for continuous metabolic data collection from subjects throughout the 24-hour cycle. While serial cortisol collection has been a staple of circadian research for many years, new biosensors now allow for continuous cortisol level tracking amongst study subjects (Kusov et al., 2023; Trusso et al., 2022). Some researchers have used machine learning to improve the efficacy of these sensors, with Shahub et al. describing how they developed a machine learning guided

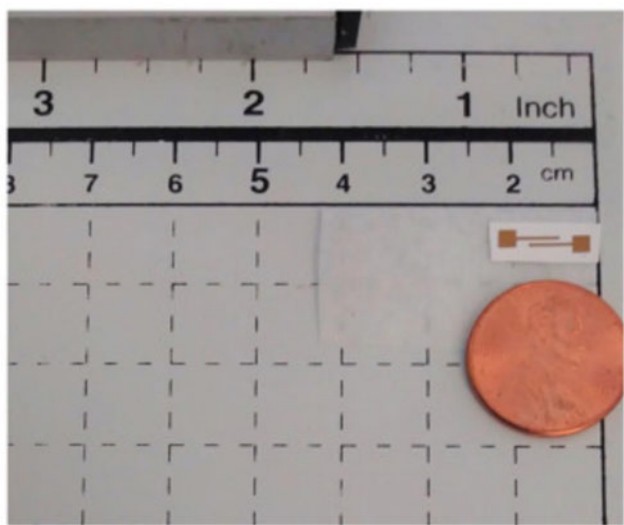

**Figure 1.** Scaled photo of machine learning-guided cortisol biosensor (Shahub et al., 2022). Image used under creative commons CC-BY license from *Sensing and Biosensing Research*.

biosensor for detecting cortisol in passive sweat (see Figure 1) (Shahub et al., 2022). Continuous glucose monitoring has also benefited from improved biosensing technology, allowing for circadian tracking of this biomarker as well (Sardesai et al., 2023; Santos-Báez et al., 2024). Core body temperature is now being incorporated into biosensing technology as well, improving the accuracy of this basic metabolic datapoint (Żmigrodzki et al., 2024). Biosensors can also continuously monitor inflammatory proteins in the sweat, providing a window into immune function over the circadian cycle (Jagannath et al., 2022).

### Real-time breath analysis

Volatile metabolites can be collected from human breath in order to obtain real-time metabolic data. *The secondary electrospray ionization machine* (SESI) is one recently developed device which allows for convenient collection of metabolites in the breath, and has been used in order to collect serial metabolite samples, thus enabling circadian rhythmic analysis (see Figure 2). (Brown and Sinues, 2021)

### Bioenergetic flux analysis

McClaren et al. describe their newly developed *metabolic flux analyzer*, a device which requires just one blood sample in order to analyze the circadian activity of mitochondria. The metabolic flux analyzer allows for the observation of mitochondrial bioenergetics and cross-correlates this data with the expression of Clock genes in white blood cells, offering a fast and minimally invasive alternative to other circadian data collection methods (McLaren et al., 2024).

### Metabolic carts and metabolic chambers

Metabolic carts and metabolic chambers are both respiratory data collection technologies used to track metabolic rate. Metabolic cart technology includes a variety of systems used to measure the resting metabolic rate of a subject by creating a closed breathing system in which the subject's minute-by-minute oxygen consumption and carbon dioxide production may be measured. Some metabolic carts position subjects beneath a ventilated hood, while others require the donning of a mouthpiece or face mask attached to the system (Chen et al., 2020). Metabolic chambers (see Figure 3)

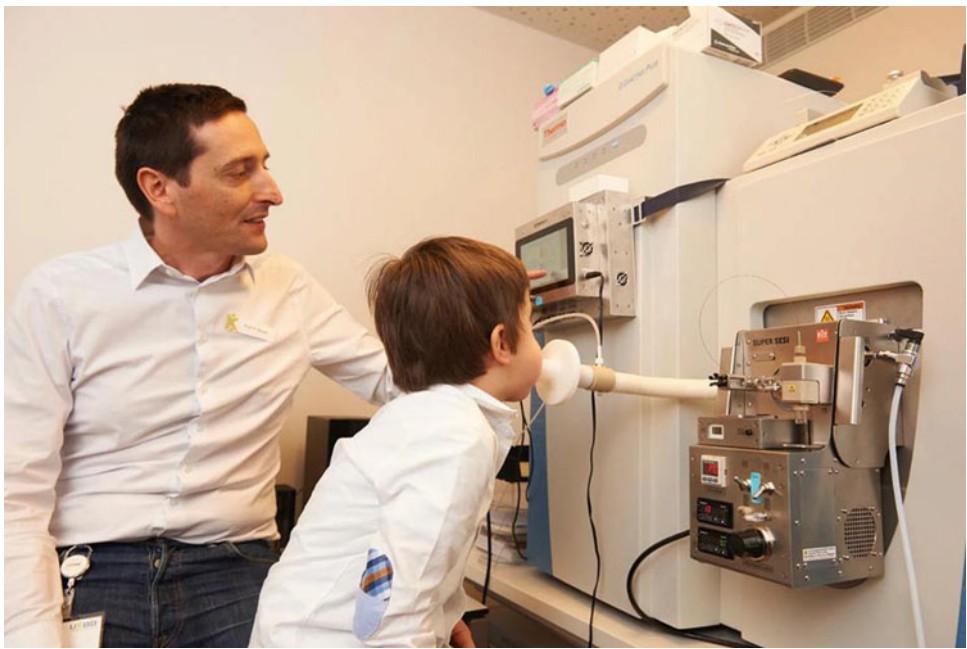

**Figure 2.** Commercially available secondary electrospray ionization machine (SESI) (Brown and Sinues, 2021). Figure used with permission from Springer Nature.

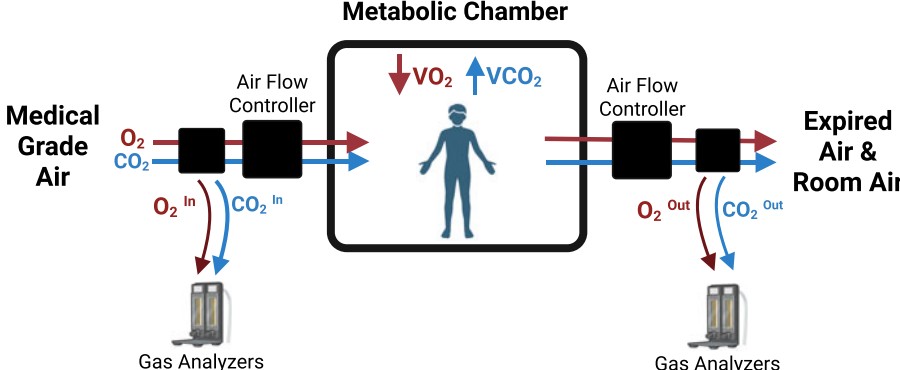

**Figure 3.** Metabolic chamber (i.e. respiratory chamber) concept diagram. Image inspired by design described by Chen et al. (Chen et al., 2018). Created and published with permission by BioRender.

(Chen et al., 2018), sometimes called respiratory chambers, are sealed, airtight rooms which serve the same purpose; a subject is placed alone in a chamber for a period of time, during which their total oxygen consumption and carbon dioxide may be measured to determine indirect calorimetry and metabolic rate over a period of time (Ravussin et al., 1986). These techniques have recently been used in various metabolic rhythm studies (Meng et al., 2020; de Wit-Verheggen et al., 2023), and could be incorporated into circadian rhythm study designs as well.

*Combination protocols*
Some recent research combines multiple methods in order to generate more complete metabolic rhythm profiles for subjects. For example, Harmsen et al utilized serial muscle biopsy (examining peripheral Clock gene expression) along with serial indirect calorimetry and serial metabolite sampling of glucose, insulin, free fatty acids, and triacylglycerol, in order to generate circadian metabolic data for their subject pool (Harmsen et al., 2024). Dineen et al. used serial cortisol and cortisone blood sampling, as well as serial adipose tissue microdialysis for cortisol

and cortisone, in order to obtain more accurate measurements of these metabolic markers over time in their study of adrenal insufficiency (Dineen et al., 2023). And Koch et al. combined actigraphy data with energy expenditure calculations, indirect calorimetry, and energy intake calculations in order to generate metabolic data for their subjects over time (Koch et al., 2025; McNeil et al., 2024).

*Peripheral melatonin biomarkers: new approaches to measurement*

As discussed previously, salivary melatonin and urinary 6-sulfatoxymelatonin are both established markers of circadian phase which can be accurate and useful, as long as the masking effect of light exposure is diligently avoided. Some recent efforts have revisited these techniques in order to improve their accuracy and better avoid the masking effect of light exposure. Murray et al. developed a protocol for at-home salivary melatonin collection for the purpose of DLMO calculation, in which they created stand-ardized instructions to avoid light masking (Murray et al., 2024).

Stone et al. examined the role of individual light sensitivity in DLMO data, and developed data analysis methods to control for this variable when determining circadian phase based on salivary melatonin (Stone et al., 2020).

Urinary 6-sulfatoxymelatonin has likewise benefited from recent analytical innovations which improve the usefulness of this biomarker. Melone et al. (2023) and Doyle et al. (2022) both describe new methods of rhythmic analysis which allow the extrapolation of continuous circadian rhythm from serial urinary 6-sulfatoxymelatonin samples. Van Faassen et al. describe a method to account for age and individual biological variation when tracking 6-sulfatoxymelatonin levels, enabling more accurate interpretation of this biomarker when inferring circadian rhythm from individual samples (van Faassen et al., 2020).

### Exploring mechanistic pathways: circadian rhythm and psychiatric phenotypes

As novel circadian measurement methods continue to emerge, circadian characteristics may be more precisely measured in subjects over time, thus allowing a more detailed study of the bidirectional mechanisms linking circadian abnormalities and psychiatric phenomena. Below we review recent longitudinal studies in which researchers have designed novel protocols to study the relationship between circadian rhythm disturbances and mood disorders.

#### *Circadian alignment and depression*

Recent studies have confirmed prior findings that circadian alignment, i.e. alignment between an individual's circadian phase and the actual timing of their sleep cycle, is positively correlated with mood. In 2023, Emens and Lewy published findings assessing circadian alignment within a group of 25 medical students without psychiatric pathology. In this population, they found that circadian misalignment, specifically a later DLMO relative to midsleep, was associated with worse mood (Emens and Lewy, 2024). Asarnow et al., recently used a combination of actigraphy, DLMO measurements, and self-report of sleep habits to assess alignment between circadian biology and actual sleep-wake behaviors in adolescents with depression symptoms; they found that as subjects' circadian alignment improved, they also experienced a reduction in depression symptoms (Asarnow et al., 2023).

#### *Chronotype stability and bipolar disorder*

Longitudinal chronotype tracking is a technique which reveals extended patterns of chronotype, i.e. an individual's preference for mornings or evenings, and how this may vary over time. In 2024, Sperry et al. analyzed chronotype patterns over several years in a pool of subjects with and without bipolar disorder, finding that a bipolar disorder diagnosis is characterized by greater chronotype instability compared to controls (Sperry et al., 2024).

#### *Delayed circadian phase and depression*

Delayed circadian phase, which refers to a later timing of various circadian timepoints compared to average values, has often been found in association with psychiatric pathology. Zhang et al. reexamined this finding using multimodal wearable sensing via Fitbit, which tracked sleep patterns, M10 onset, and heart rate acrophase in a group of subjects over two years. Their findings were consistent with previous studies showing that delayed circadian rhythms were associated with increased depression symptoms (Zhang et al., 2024).

#### *Sleep variability and depression*

Matcham et al. used Fitbit to track sleep patterns and depression symptoms in their subjects; they found that greater sleep variability was predictive of major depressive disorder (MDD) relapse amongst their participants (Matcham et al., 2024).

### Novel device-guided circadian therapeutics

Therapies targeting circadian disturbance have often proved beneficial in the treatment of psychiatric disorders, with patients seeing total psychiatric improvements beyond the resolution of their isolated sleep concerns. Given an increasing awareness of the mechanistic relationships between circadian rhythm disturbance and psychiatric pathology as a whole, as well as the development of new devices which manipulate these mechanisms, device-guided circadian-targeted therapies may become increasingly utilized by clinicians. Examples of such new technologies are described below.

#### *Near-infrared transcranial photobiostimulation*

Near-infrared transcranial photobiostimulation has gained momentum in recent years as a noninvasive technique which improves mitochondrial function in the brain, with measurable benefits for neurodegenerative disorders such as Alzheimer's disease (Zomorrodi et al., 2019). More recently, this technique has been shown to produce antidepressant and hypnotic effects which may be useful for the treatment of SCRD and mood disorders. In 2024, Guu et al. adapted this technology into a wearable headband which could administer treatments at home. Participants reported improved sleep quality via the Pittsburgh Sleep Quality Index, which persisted for the 12 weeks of the study (Guu et al., 2025).

#### *Parcel-guided transcranial magnetic stimulation (TMS)*

Recent studies have examined the use of transcranial magnetic stimulation (TMS) in treating SCRD. Tang et al. developed a protocol in which TMS targets were selected on an individual basis, dependent on functional connectivity network data from individual study participants. Anomalies in individual connectivity networks were detected using a machine-learning connectivity software, and then translated into TMS targets with the goal of modifying participants' central executive, salience, and default mode networks. This individualized approach produced favorable results, with participants showing improvements in Pittsburgh Sleep Quality index scores as well as improvements in depression and anxiety symptoms (Tang et al., 2024).

### Conclusion

The relationship between sleep and circadian disturbances and mood disorders has been explored through decades of research which shows clear associations between SCRD, negative mood states, and the occurrence of psychiatric pathology. Therapies targeting sleep and circadian symptoms often foster an incidental improvement of mood disorder symptoms as well, further suggesting a causal relationship between the two phenomena. However, our understanding of this causal relationship has remained limited by the scarcity of tools traditionally available for studying circadian rhythm fluctuations in human subjects using precise biomarkers. A review of the field suggests that many new techniques have emerged in recent years which will allow more precise collection of circadian data as it appears in gene

expression patterns, metabolic patterns, and other biomarkers; however, these techniques have yet to be widely adopted by researchers conducting longitudinal studies in human subjects. At this time, the field of sleep and circadian disturbances and mood disorders offers rich opportunity for those looking to implement new techniques in the examination of these two closely related phenomena.

**Data availability statement.** Data availability is not applicable to this article as no new data were created or analyzed in this study.

**Author contributions.** K.L. wrote the manuscript; C.M. edited the manuscript.

**Financial support.** Work from our lab was funded by a Physician-Scientist Institutional Award from the Burroughs Wellcome Fund to K.L. as well as the Baszucki Brain Research fund, National Institute of Mental Health (NIMH) (MH106460;MH111601), National Institute on Drug Abuse (NIDA) (DA039865;DA046346), and the WoodNext Foundation to C.M.

**Competing interests.** The authors have no relevant financial or non-financial interests to disclose.

**Ethical approval.** Ethical approval and consent are not relevant to this article type.

## Connections references

**Hickie IB, McCarthy MJ, Crouse JJ, and Carpenter JS**. Are SCRD the cause or simply the consequence of depression or other mood disorder sub-types? *Research Directions: Depression.* 2024;**1**:e1. https://doi.org/10.1017/dep.2023.18.

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
