## [Reviewer Report]

1. Overall impressions: This paper is well written, and presents a range of emerging measurement approaches in the domain of circadian measurement in mood disorders. It is well-referenced with contemporary citations, and will be of interest to readers.

2. Structure and content coverage: The paper has two stated aims - to review new measurement approaches, and present evidence of a causal role of SCRD in psychiatric conditions (actually mood disorders) and endophenotypes (this doesn't actually appear). The paper's content does include these two topics, but also includes a brief review of the current state of circadian therapeutics (behavioural and pharmacological). This content coverage is common in the circadian-mood field ("let's say something about everything, from measurement to mechanisms to treatments"), but can tend to lead to a relatively superficial coverage. I wonder if the paper would be stronger without the treatment section (except perhaps the device-guided techniques, which continues the theme of Section II and III)? Seems to me that the cross-cutting theme in all sections is the role of new measurement approaches â€“ this could be made clearer, and may help tighten each section.

3. Section II is a strength of the paper, with a number of striking innovations of which this reviewer was unaware. Given the title of this paper, it would be useful to add some consideration of the strengths and weakness of these inventions in relation to utility in mood disorder research specifically. It might also be worthwhile making it clear to the reader that this section (and indeed the paper as a whole) is adopting an uncritical approach to these emerging developments â€“ the paper is about potential advances, not proven developments.

3. Section III: Interesting material on novel protocols for investigating associations between circadian function and mood disorders. An observation: â€˜Causeâ€™ is a debated construct in biology, especially when weâ€™re considering the interplay between different nominal nodes in a single complex system (e.g., circadian node and mood regulation node) [1]. There is also consensus that the relationship between circadian function and mood disorders will be bidirectional. And most (but not all) of the examples given here are cross-sectional correlations. There is some causal evidence using some interesting approaches [2,3,4]

4. Section IV: The strength of this section is the focus on new developments. The material on existing treatments (CBT-I, IPSRT) is less interesting and lacks the critical approach we expect to see when such existing treatments (as opposed to treatments earlier in the translational pipeline) are being discussed. It might be worth noting for the behavioural treatments that we actually have little evidence that they operate through circadian mechanisms? Evidence that sleep change might be a mechanism is not the same thing as evidence that circadian change might be a mechanism, so the term SCRD is not ideal when talking about mechanisms and causes. Sleep problems are explicitly constitutive of mood disorders, while circadian abnormalities may be more separable and thus causal.

[1] Bechtel, W., Circadian Rhythms and Mood Disorders: Are the Phenomena and Mechanisms Causally Related? Frontiers in Psychiatry, 2015. Volume 6 - 2015.

[2] Song, Y.M., et al., Causal dynamics of sleep, circadian rhythm, and mood symptoms in patients with major depression and bipolar disorder: insights from longitudinal wearable device data. EBioMedicine, 2024. 103: p. 105094.

[3] Zhou, F., et al., Assessing the causal associations of insomnia with depressive symptoms and subjective well-being: a bidirectional Mendelian randomization study. Sleep Medicine, 2021. 87: p. 85-91.

[4] Cai, L., et al., Causal links between major depressive disorder and insomnia: A Mendelian randomisation study. Gene, 2021. 768: p. 145271.